# Factors influencing estimates of HIV-1 infection timing using BEAST

**Bethany Dearlove**[1,2], **Sodsai Tovanabutra**[1,2], **Christopher L. Owen**[1,2], **Eric Lewitus**[1,2], **Yifan Li**[1,2], **Eric Sanders-Buell**[1,2], **Meera Bose**[1,2], **Anne-Marie O'Sullivan**[1,2], **Gustavo Kijak**[1,2], **Shana Miller**[1,2], **Kultida Poltavee**[1,2], **Jenica Lee**[1,2], **Lydia Bonar**[1,2], **Elizabeth Harbolick**[1,2], **Bahar Ahani**[1,2], **Phuc Pham**[1,2], **Hannah Kibuuka**[3], **Lucas Maganga**[4], **Sorachai Nitayaphan**[5], **Fred K. Sawe**[6], **Jerome H. Kim**[7], **Leigh Anne Eller**[1,2], **Sandhya Vasan**[1,2], **Robert Gramzinski**[1], **Nelson L. Michael**[8], **Merlin L. Robb**[1,2], **Morgane Rolland**[1,2]\*, **the RV217 Study Team**[¶]

1 U.S. Military HIV Research Program, Walter Reed Army Institute of Research, Silver Spring, Maryland, United States of America, 2 Henry M. Jackson Foundation for the Advancement of Military Medicine, Inc., Bethesda, Maryland, United States of America, 3 Makerere University Walter Reed Project, Kampala, Uganda, 4 National Institute for Medical Research-Mbeya Medical Research Centre, Mbeya, Tanzania, 5 Armed Forces Research Institute of Medical Sciences, Bangkok, Thailand, 6 Kenya Medical Research Institute/U.S. Army Medical Research Directorate-Africa/Kenya-Henry Jackson Foundation MRI, Kericho, Kenya, 7 International Vaccine Institute, Seoul, S. Korea, 8 Center for Infectious Diseases Research, Walter Reed Army Institute of Research, Silver Spring, Maryland, United States of America

¶ Membership of RV217 Study Team is provided in the acknowledgments.
\* mrolland@hivresearch.org

**Data Availability Statement:** Sequences are available on GenBank under accession numbers KY580473—KY580727 and MN791130—MN792579.

## Abstract

While large datasets of HIV-1 sequences are increasingly being generated, many studies rely on a single gene or fragment of the genome and few comparative studies across genes have been done. We performed genome-based and gene-specific Bayesian phylogenetic analyses to investigate how certain factors impact estimates of the infection dates in an acute HIV-1 infection cohort, RV217. In this cohort, HIV-1 diagnosis corresponded to the first RNA positive test and occurred a median of four days after the last negative test, allowing us to compare timing estimates using BEAST to a narrow window of infection. We analyzed HIV-1 sequences sampled one week, one month and six months after HIV-1 diagnosis in 39 individuals. We found that shared diversity and temporal signal was limited in acute infection, and insufficient to allow timing inferences in the shortest HIV-1 genes, thus dated phylogenies were primarily analyzed for *env*, *gag*, *pol* and near full-length genomes. There was no one best-fitting model across participants and genes, though relaxed molecular clocks (73% of best-fitting models) and the Bayesian skyline (49%) tended to be favored. For infections with single founders, the infection date was estimated to be around one week pre-diagnosis for *env* (IQR: 3–9 days) and *gag* (IQR: 5–9 days), whilst the genome placed it at a median of 10 days (IQR: 4–19). Multiply-founded infections proved problematic to date. Our ability to compare timing inferences to precise estimates of HIV-1 infection (within a week) highlights that molecular dating methods can be applied to within-host datasets from early infection. Nonetheless, our results also suggest caution when using uniform clock and population models or short genes with limited information content.

**Funding:** This work was supported by a cooperative agreement between The Henry M. Jackson Foundation for the Advancement of Military Medicine, Inc., and the U.S. Department of the Army [W81XWH-07-2-0067 (NLM), W81XWH-11-2-0174 (NLM), W81XWH-18-2-0040 (RG)]. The views expressed are those of the authors and should not be construed to represent the positions of the U.S. Army, the Department of Defense, or the Department of Health and Human Services. The funders had no role in study design, data collection and analysis, decision to publish, or preparation of the manuscript.

**Competing interests:** The authors have declared that no competing interests exist.

## Author summary

Molecular dating using phylogenetics allows us to estimate the date of an infection from time-stamped within-host sequences alone. There are large datasets of HIV-1 sequences, but genome and gene analyses are not often performed in parallel and rarely with the possibility to compare results against a known narrow window of infection. We showed that all but the longest genes are near-clonal in acute infection, with little information for dating purposes. For infections with single founders, we estimated the eclipse phase—the time between HIV-1 exposure and the first positive diagnostic test—to last between one and two weeks using *env*, *gag*, *pol* and near full-length genomes. This approach could be used to narrow the date of suspected infection in ongoing clinical trials for the prevention of HIV-1 infection.

## Introduction

The exact age of a human immunodeficiency virus (HIV-1) infection is rarely known, as there is a lag between the time of infection and diagnosis. However, knowing when infection occurred is important for assessing incidence rates, transmission dynamics and preventive interventions. Fiebig stages are used to define the progression of the early stages of clinical infection, using the time until laboratory tests for HIV-1 antigens and HIV-1-specific antibodies become positive [1]. Fiebig stage I refers to the earliest time that the virus can be identified in the bloodstream, by viral RNA test using PCR. Preceding this stage is the eclipse phase, corresponding to the interval between initial exposure and HIV-1 detection, during which replication occurs first near the site of HIV-1 transmission. Previously, studies using serologic and RNA tests on plasma serially-sampled from blood bank donors estimated the eclipse phase between 3 and 33 days, with several reports emphasizing a window of ten to twelve days [1–5]. Besides diagnostic assays, HIV-1 sequences derived from infected participants can determine the age of an infection. Using the within-host sequence diversity that accumulates through error-prone replication after infection, it is possible to reconstruct the evolutionary history of a viral infection within an individual. Previous work has shown that polymorphic nucleotides in the *pol* region can be used to identify recent infection events within one year [6], and the precision of these estimates has improved with deep-sequencing efforts [7]. We recently showed that it is possible to estimate the date of infection with a resolution on the scale of days using a cohort with a known narrow window of infection, the RV217 cohort [8,9]. This cohort enrolled more than 3,000 seronegative high-risk individuals in four countries (Kenya, Tanzania, Thailand and Uganda) for twice-weekly HIV-1 RNA tests and identified 155 acute infections, thereby providing a unique opportunity to compare dating methods against real data, as the tight bounds on the time between the last negative and first positive test can be used to evaluate the accuracy of molecular dating methods. Diagnosis, or Day 0, was defined as the first sample which was reactive for HIV-1 RNA. Near full-length (NFL) genome sequences were obtained for a subset of 39 acutely infected participants, including 10 men, 5 transgender women and 1 woman from Thailand, and 23 women from East Africa.

We previously analyzed *env* sequences to estimate the eclipse phase using the time to most recent common ancestor (TMRCA) of the within-host phylogeny of sequences from the first six months of infection [8]. Here, we use the same methodology to compare estimates of the date of infection for the same study participants using individual genes and NFL genomes. Large numbers of HIV-1 sequences are publicly available (>850,000), yet few studies have

evaluated how findings can differ based on genome- or gene-specific analyses. We evaluated how certain parameters influenced the results of our BEAST analyses and identified best-practice approaches for minimizing their effect. We showed that genetic and temporal signal, model choice and founder multiplicity greatly affected timing estimates. Whilst estimation of the date of infection remained problematic for participants with infections founded by multiple variants, the estimated date of infection for participants with single founder infections tended to fall within one to two weeks prior to diagnosis when estimated using the NFL genome, *env*, *pol* and *gag*.

## Results

### Characterization of infections with single and multiple founder variants

We analyzed HIV-1 genomes from 39 participants sampled longitudinally at three time points after the initial detection of HIV-1 viremia by a positive HIV-1 RNA test: at approximately one week (median: 4 days), one month (median: 32 days) and six months (median: 170 days); this diagnosis occurred a median of four days after the last negative test [8]. Approximately ten NFL genomes were derived via endpoint-dilution from plasma samples for each participant at each time point, yielding about thirty NFL genomes per participant (total = 761 NFL genomes, 379 5' half genomes, 416 3' half genomes and 81 additional *env* sequences). In East Africa (n = 23 women), the infections were most commonly identified as subtype A1 (n = 9), followed by subtype C (n = 3) and various A1/D recombinants. In Thailand (n = 10 men, 5 transgender women and 1 woman), all but two of the infections were CRF01_AE (n = 14); the remaining infections were subtype B (male participant) and a B/CRF01_AE recombinant (transgender participant).

Infections were classified into single or multiple founders using the genome sequences across all time points and the current standard approach: visual inspection of sequence alignments, Highlighter plots and phylogenetic trees reconstructed in IQ-TREE, and quantitative measures of intra-host diversity including the maximum pairwise diversity and ratio of shared to private mutations [8,10]. Additionally, we used a relatively new metric, the principle eigenvalue of the modified graph Laplacian (MGL), which summarizes the diversification patterns of phylogenetic trees and has been shown to positively correlate with the multiplicity of infection [8,11].

The majority of infections were founded by a single population (n = 28), with 11 infections founded by two or more distinct (but still closely related) lineages of variants. One week after diagnosis, the number of polymorphic sites was ten-fold higher in multiple founders (median: 136, IQR: 52.5–171) compared to those with single founders (median: 13, IQR: 11–16.5) for the NFL genome, with a two-fold difference still seen at six months (Fig 1A and 1B). Infections with multiple founders by definition had more phylogenetically informative sites (polymorphisms found in at least two sequences), due to the relatively deep branching events between founder variants (Fig 1C and 1D).

The single versus multiple founders classification above was obtained using NFL genomes and we wanted to investigate if one of the methods used, the principle eigenvalue of the modified graph Laplacian (MGL), could yield similar results across the different genes despite the low number of polymorphic sites in some genes (Fig 2). To define a cut off between the two subsets, we used the median, jump and partition criteria; all three gave consistent results for NFL genomes. Though results were broadly consistent, there was a discrepancy between the three criteria for four borderline cases (participant 20225 in *env*, 40061 in *nef*, 30112 in *tat* and 40112 in *vpu*). This was due to the distribution of the data, with the threshold effectively rounded up to the value of the $\lambda^*$ at the threshold [11]. We also found discrepancies between

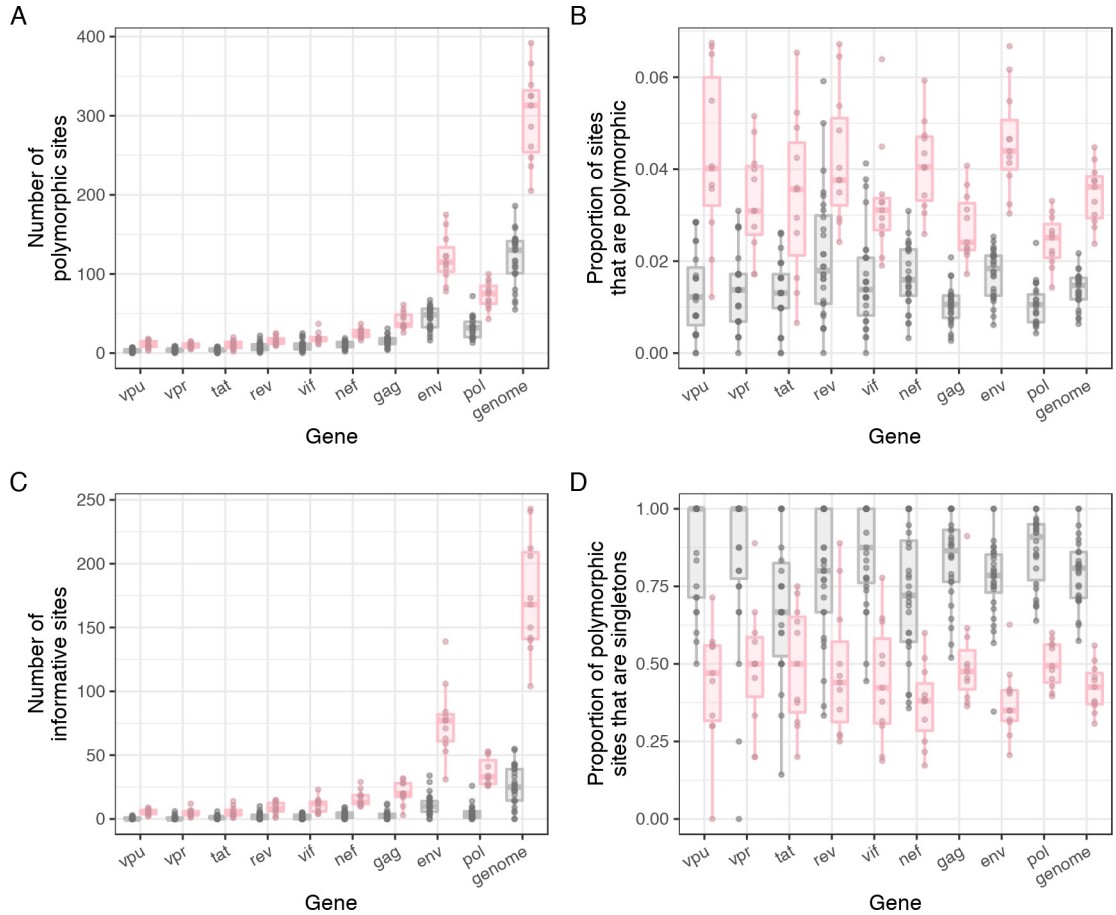

**Fig 1. Informative sites identified in the first six months of infection across the 9 HIV-1 genes and genome for 39 participants in the RV217 cohort.** A) The number of polymorphic sites. B) The proportion of polymorphic sites. C) The number of informative sites. D) The proportion of polymorphic sites that were only found in one sequence. Genes are ordered by median sequence length across participants. Points are colored gray for participants with infections founded by a single variant, and pink for those founded by multiple variants.

genes. For example, participant 10220, who had an infection with multiple founders, was classified with the single founders for *env*, *tat* and *rev*, where the sequences were more homogeneous than in the rest of the genome. Likewise, the infection in participant 40265 corresponded to a single founder but was misclassified as a multiple founder for five out of nine genes.

As we previously showed that estimates of the date of infection were poor for participants with multiple founders [8], we will focus on participants infected by a single founder for the majority of this paper. In the last section, we compare estimates of the eclipse phase using NFL genomes for multiple founders, including after splitting the sequences into founder subpopulations.

## Low genetic diversity in the first month of infection for single founders

HIV-1 is measurably evolving, even over relatively short timescales. However, as we have seen, the information content varies across the genome and by participant. Not all HIV-1 genes were equally informative in the first six months of infection, and diversity was generally low in sequences from participants with single founders (Fig 1). The number of polymorphic sites

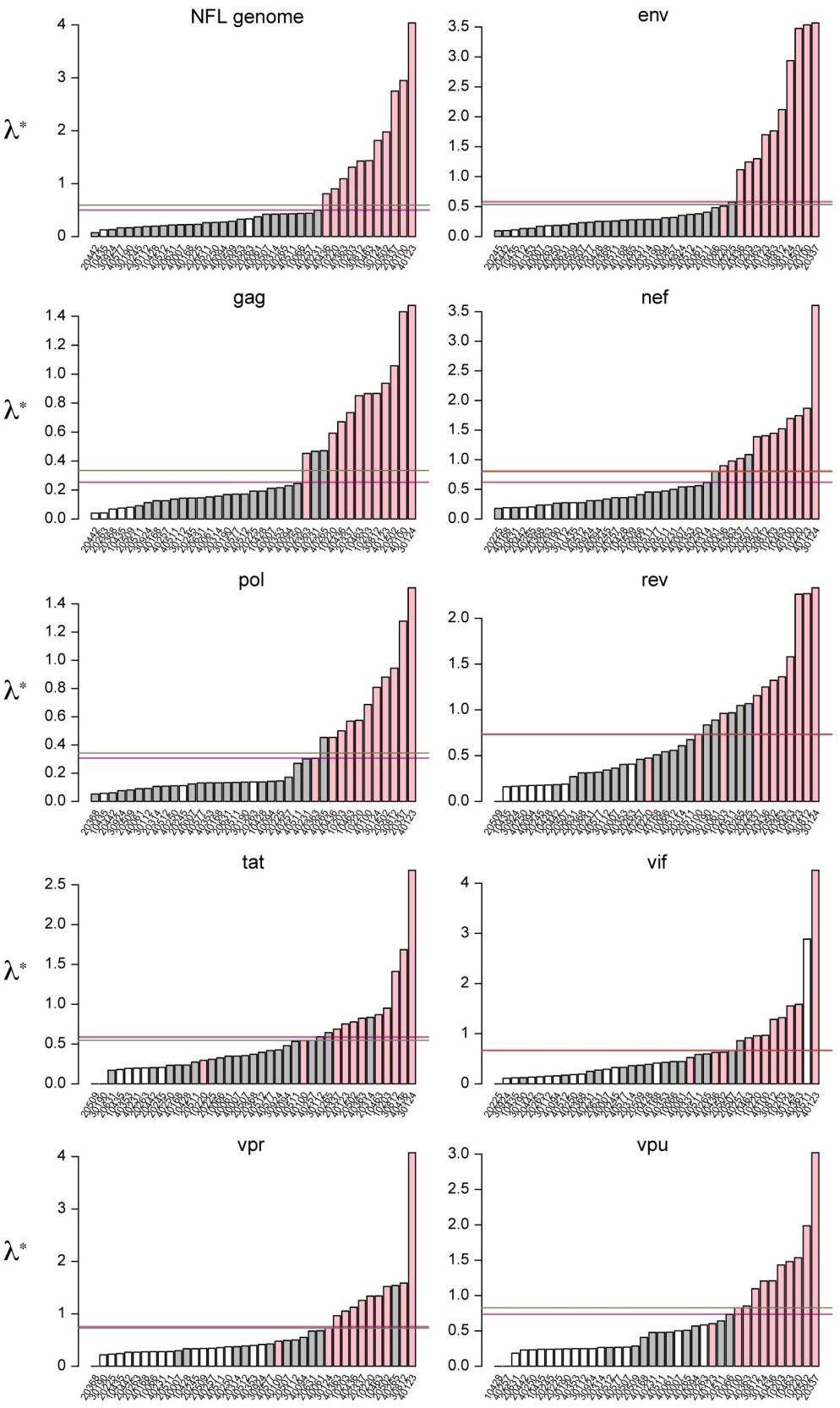

**Fig 2. Identification of infections with single versus multiple HIV-1 founders.** The principal eigenvalues from the modified graph Laplacian are compared for each participant and gene. Barplots are sorted in increasing order, and values shifted so that the smallest value is zero. Bars are colored according to whether participants were classified as a single founder (gray) or multiple founder (pink) with NFL genomes. White bars show sequence datasets in which there were no informative sites. Lines indicating thresholds inferred from the median (brown), jump (blue), and partition (red) criteria of the principal eigenvalue test of founder multiplicity are shown.

increased with the length of the gene; the median number of polymorphic sites was ≤10 for all genes shorter than *gag* (*vpu*: 3; *vpr*: 4; *tat*: 4; *rev*: 6.5; *vif*: 8; *nef*: 10), with *env* showing the highest diversity (median: 48) (Fig 1A). Fig 1B shows that the proportion of polymorphic sites was generally consistent between genes (overall median: 0.013, IQR: 0.0087–0.0188), though tended to be higher in *env* (median: 0.018, IQR: 0.013–0.021) and one of the genes overlapping with it, *rev* (median: 0.018, IQR: 0.011–0.03). The majority of mutations in single founder infections were singletons. The median percentage of polymorphic sites across participants found in only a single sequence ranged from 67% in *tat* to 100% in *vpu* and *vpr* (Fig 1D). Nearly two thirds of participants had no informative sites in *vpu* and *vpr* (63% for *vpu* and 61% for *vpr* respectively). This proportion was higher when only considering the first month since diagnosis (the first two time points), as shared mutations were rare even at the NFL genome level (median: 2, IQR: 1–4); the percentage of participants with no informative sites at the second sampling time point was greater than 60% for all genes except *env* (46%) and NFL genome (15%) (S1 Fig). Two participants, 10435 and 20263, had no informative sites in the NFL genomes after sixth months (NFL genome median for single founders: 25, IQR: 14–39). These two participants also had some of the lowest number of polymorphic sites over the first 6 months of infection, 55 and 66 respectively, compared to a median of 130 sites in the NFL genome for infections with single HIV-1 founders.

For accurate estimation of the root of our within-participant phylogenies using phylogenetic reconstruction in BEAST, it is important that there is sufficient information in the sequences [12,13]. A first step was to exclude participant datasets in which there was insufficient genetic signal, i.e. those that contained no informative sites (all the mutations were singletons). We also tested for temporal signal, by regressing the root-to-tip distance calculated from phylogenies reconstructed in IQ-TREE against the sampling times for each within-participant sequence set. Datasets without a significant, positive slope were removed from the phylogenetic analyses in BEAST (S2 Fig). This resulted in the exclusion of participant 20442 for *env*, *tat* and *rev*; 20245 for *env* and *nef*; and 20314 (*vpr*) and 40265 (*vpu*). Additionally, participant 20368 was removed from the NFL genome analysis as no full length sequences were available, only half sequences from either the 3' or 5' end with 1293 sites overlap. This left between 9 (*vpu*) and 25 (NFL genome) single founder participant datasets to analyze in BEAST.

## Estimates of eclipse phase for single founders were between one and two weeks

We used the phylogenetic software BEAST v1.8.3 to implement 16 combinations of clock and population model for each participant, considering each gene individually as well as the NFL genome for comparison purposes [14]. We took the median time to most recent common ancestor (TMRCA) of the best-fitting model (selected using the highest marginal likelihood estimated by stepping-stone sampling) as a point estimate for the date of infection (Fig 3).

Across infections with single founders, the median point estimate for the date of infection was 6 days before diagnosis for *gag* and *env* (*gag* IQR: 4–9; *env* IQR: 2–9). The estimates of the date of infection for the NFL genome and *pol* fell a little earlier relative to diagnosis: 10 days (IQR: 5–20) for the whole genome, and 17 days (IQR: 5–40) for *pol* (S1 Table). Overall, most

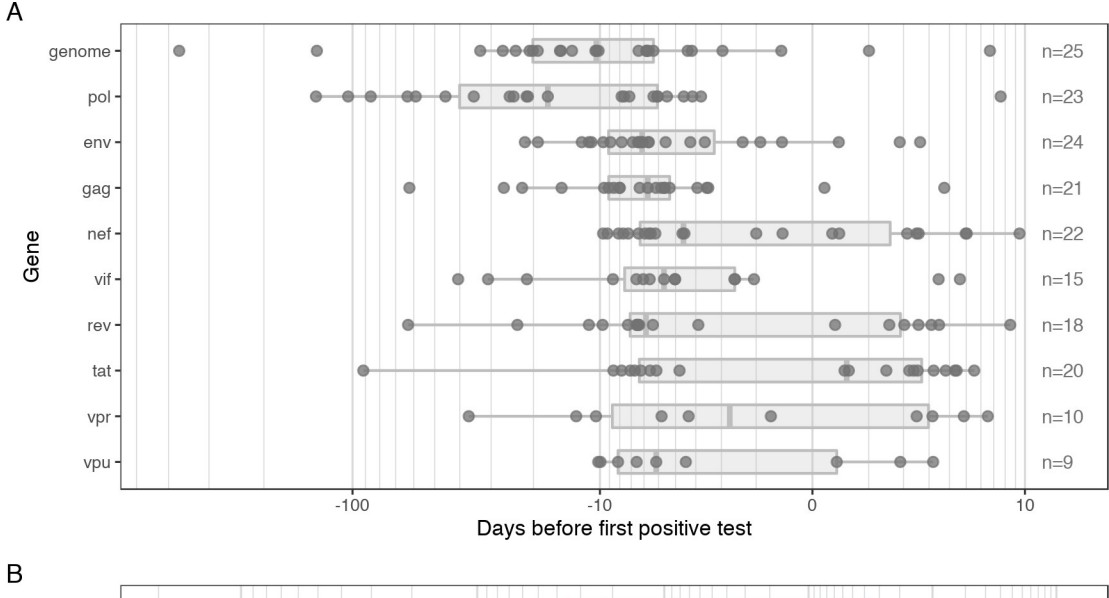

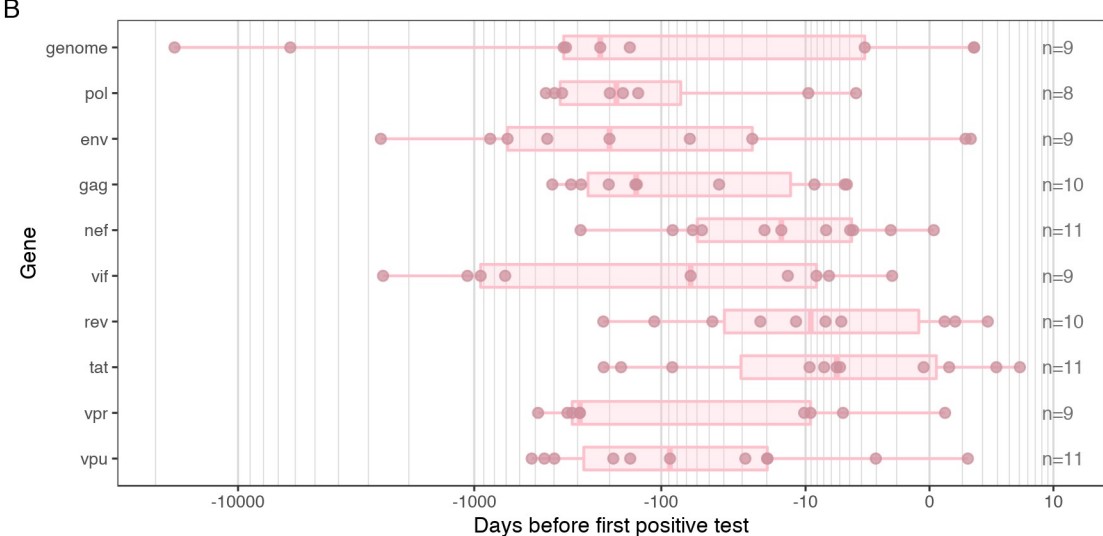

**Fig 3. Estimates of the date of infection by gene and founder type.** Points are colored gray for participants with infections founded by a single variant (A), and pink for those founded by multiple variants (B). The scale is shown with a power modulus transformation for visibility, and is different for infections with single or multiple HIV-1 founders.

of the participants had estimates for the dates of infection that fell in the two weeks prior to diagnosis: 79% for *env*, 71% for *gag*, 52% for NFL genomes and 43% for *pol* (Fig 3A and S1 Table). Shorter genes tended to estimate a date of infection much closer to diagnosis, with the IQR for *nef*, *rev*, *vpr* and *tat* encompassing estimates of infection date after diagnosis (36%, 39%, 40% and 55% of participants respectively) (S1 Table and S3 Fig).

When comparing the three longest genes and NFL genome, estimates for the same participant were not necessarily consistent (Fig 4). Curiously, for trees reconstructed from *pol*, and to some extent NFL genome, sequences seem to overestimate the eclipse phase compared to *gag* and *env* (Figs 4 and S4A). Particularly striking were participants 20442 and 20245, with estimates for the date of infection for the NFL genome more than 100 days prior to diagnosis (20442: 371 days prior to diagnosis, 20245: 133 days). These two Kenyan participants quickly controlled viremia after peak viral loads of 6.54 and 6.25 log copies/ml for 20442 and 20245

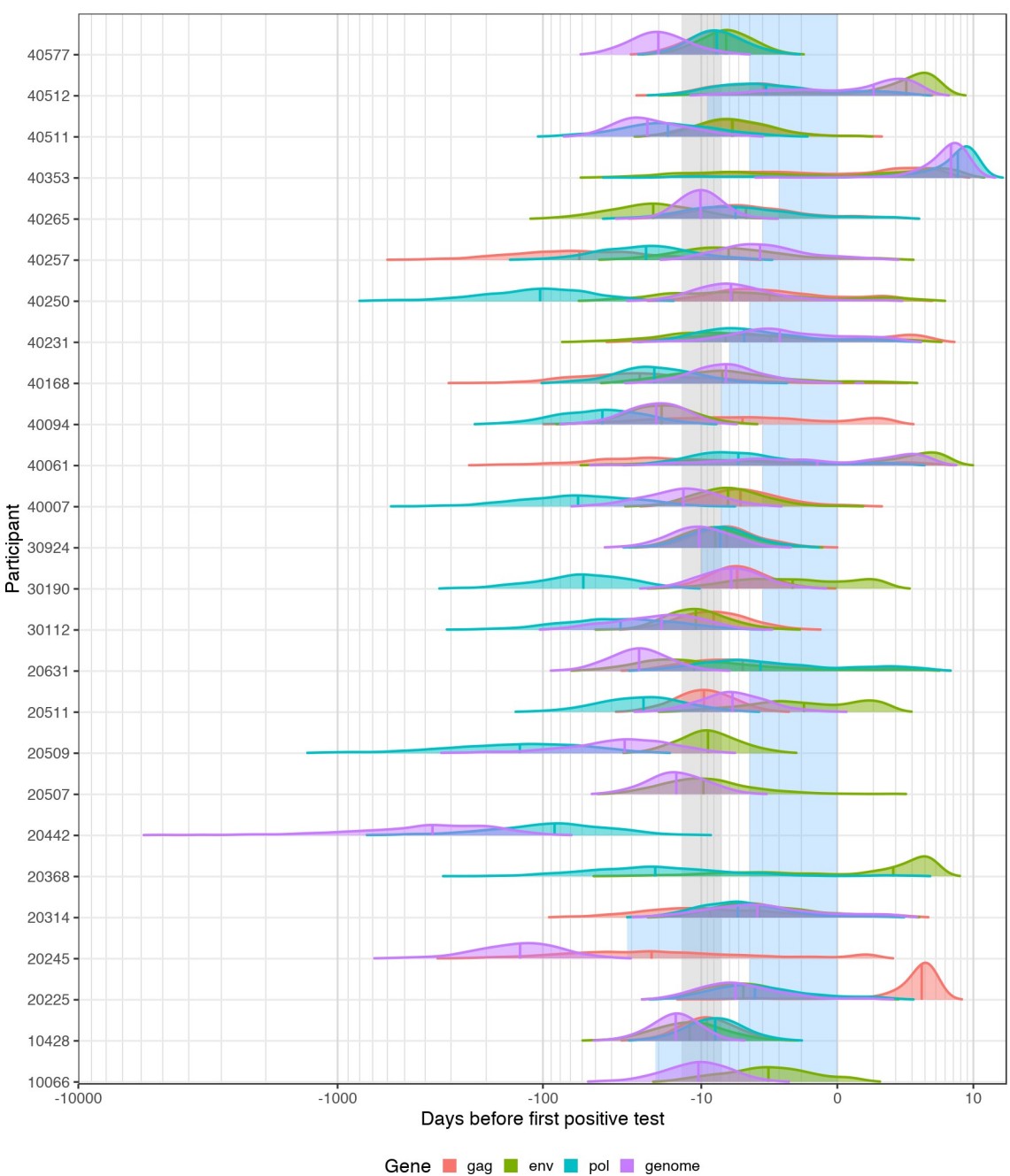

**Fig 4. Posterior distributions of the date of infection for participants with a single founder population.** Vertical lines mark the median. The shaded blue area corresponds to the interval between the last negative and first positive HIV-1 RNA test (or diagnosis date). The shaded gray rectangle highlights the period between 7 and 14 days before diagnosis.

respectively, and only had 5 (participant 20442) and 6 (participant 20245) informative sites across the three time points for the NFL genome (indeed, only the NFL genome and one other gene had sufficient genetic and temporal signal to be included in the BEAST analysis). The lack of substitutions led to very low evolutionary rate estimates (20442: $6.66 \times 10^{-7}$ subs/site/day; 20245: $2.64 \times 10^{-6}$ subs/site/day; overall median: $1.45 \times 10^{-5}$ subs/site/day), pushing the root of the phylogenies back in time.

We used the overlap coefficient, which ranges from 0 (no overlap) to 1 (complete overlap), to capture the overlap in posterior distributions between genes for each participant [15,16]. Coefficients varied widely overall but tended to show more overlap between posterior distributions for *env* and *gag*, and *env* and NFL genome (S4B Fig).

## The best-fitting model combination varied across participants and genes

The best-fitting model combination, as chosen by the highest marginal likelihood estimated using stepping-stone sampling [17], varied across both participants and genes (Fig 5). For no single participant did all 16 models run and converge. The random local clock (RLC) was particularly poor, only fitted for *env*, *pol*, *gag* and the genome. Even so, it was never fitted with the exponential population model, and in all but one case there was substantial evidence (Bayes factor>3.2 [18]) towards rejecting models fit with the RLC in favor of the best-fitting model (S5 Fig). The exception was participant 20509, with a BF of 2.1 in favor of the best-fitting model (UCLD-skyline) relative to the second best-fitting (RLC-skyline) model.

For the NFL genome, the most frequently selected population model was the skyline, selected as the best-fitting for all but one of the participants, in combination with the strict clock for 16 participants, the uncorrelated lognormal relaxed clock (UCLD) for 5 participants, and the uncorrelated exponential relaxed clock (UCED) for 3 participants. The remaining participant was fitted with the UCLD clock and birth-death population model. All participants had at least one model combination including a skyline in the top three ranked models, while 14 out of 25 participants had the skyline population model for all of the top three models selected. In addition to participant 20509 mentioned above, there were four other participants with little evidence separating the top two model combinations, with the top two models both involving the skyline: for participants 20314 and 30190, the UCLD was slightly favored over the strict clock (BF = 2 and 1.2 respectively), for participant 10066 the UCED was slightly favored over the strict clock (BF = 1.8), and for participant 40007 the strict clock was slightly favored over the UCLD relaxed clock (BF = 1.2) (S5 Fig).

The skyline was also the best population model for *pol*, *env* and *vpr*, though it was selected less frequently (61%, 58% and 40% of participants respectively). Indeed, for *env* the best single model combination was equally likely to be the skyline with a strict clock, or the UCLD relaxed clock and birth death model (n = 8). For the remaining genes, simpler population models were favored with either of the relaxed clocks: the UCLD relaxed clock with constant population for *tat* and *rev* and the UCED and birth-death model for *vpu*, *rev*, *vif*, *nef* and *gag*.

When considering the best-fitting population model, we found that the strict clock tended to estimate a longer time between infection and diagnosis compared to other clock models when in combination with the best population model (S6 Fig). For NFL genomes, the strict clock was estimated to be a median of 2.7 (IQR: 1–12) days prior to estimates using other clocks; strict clock estimates also tended to be more consistent with UCLD relaxed clock (median difference: 1.2, IQR: 0.41–4.8) than the UCED relaxed clock (median difference: 5.7, IQR: 2.2–13). Similar patterns were seen across the other genes, with the strict clock giving estimates 2.5 (IQR: 1.8–5) days prior to other clock models for *gag*, 3.4 (IQR: 0.93–7.2) for *env*, and 3.4 (IQR: 0.93–7.2) for *pol*.

## 'Standard' models tended to fit worse and gave poorer estimates than best-fitting models

We showed that although the random local clock model was clearly inferior, there was no single best-fitting combination of clock and population model for estimating the date of infection across all genes and participants. We considered how our estimates of the date of infection

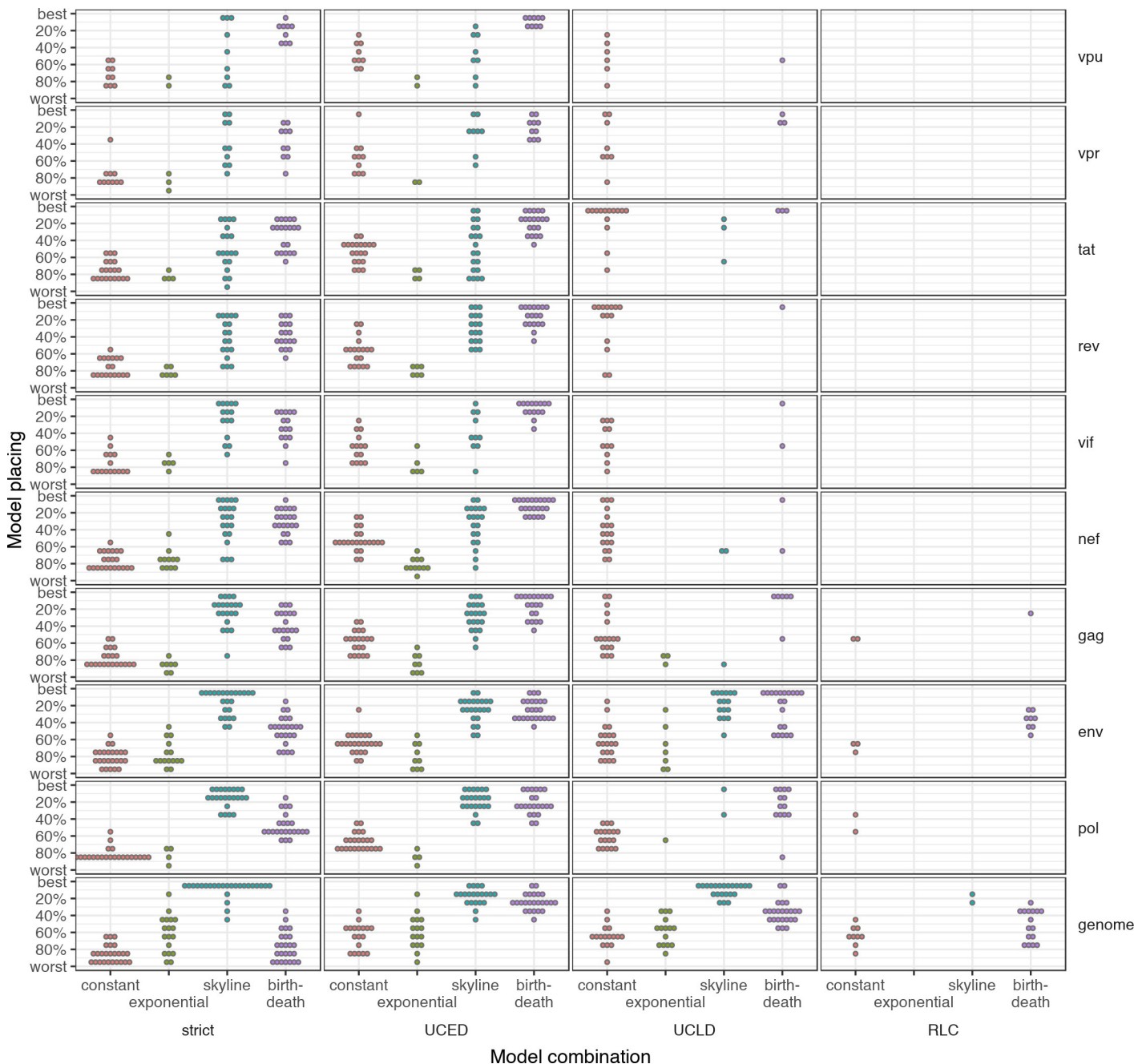

**Fig 5. Comparison of model rankings for each model combination for single founders for each gene and NFL genome.** Each combination of the four clock (strict, uncorrelated exponential (UCED) relaxed, uncorrelated lognormal relaxed (UCLD), and random local (RLC)) and four population models (constant, exponential, skyline and birth-death) are represented on the x-axis, and the model placing on the y-axis. Each dot represents the model fitted for one participant. Models were ranked by their estimated marginal likelihood and rankings scaled by the total number of models fitted for that participant and gene.

would have changed for single founders, had we only fitted one of the most commonly fitted 'standard' models: the strict clock with constant population size (the simplest possible model), and the uncorrelated log-normal relaxed clock and skyline model (allowing for most flexibility).

When the UCLD relaxed clock with skyline was fitted, it tended to be ranked highly (Fig 5); it was the top fitted model for 20% of participants for the NFL genome. However, it often had

convergence problems—in no gene or NFL genome did it fit for every participant, and even for *gag* and *pol* it was only fitted for one and two participants respectively. In contrast, the strict-constant model was fitted for all participants, but tended to rank much more lowly—for no participant or informative gene/NFL genome was it chosen as the best model, and only one participant, 40257, had it ranked within the top five for both *gag* and *pol*. Estimates for the eclipse phase were also much longer for the strict-constant model, with a median estimate across participants of 27 days for *gag*, 41 days for *pol*, 33 days for *env*, and 34 days for the NFL genome (Fig 6).

## Splitting multiple founders into founding subpopulations gave more realistic, but not robust, results

We previously showed that phylogenetic methods revealed discordant dating estimates for infections with multiple founders that reflected the age of the infection in the transmitter rather than the current host. We also showed that our limited number of sequences did not allow us to accurately resolve infections into founder variants for estimating the date of infection [8]. Here, point estimates (calculated as the median) from the best-fitting model were also wide-ranging for multiple founders when considering all sequences, from 17,127 days prior to diagnosis to 5.1 days post-diagnosis (Fig 3B). The estimates for multiple founders tended to be significantly higher than those for single founders (Wilcoxon rank sum test: p < 0.05 for all genes except *tat*, *rev* and *genome*), with the median estimated date of infection being 225 days (IQR: 3–356) for the NFL genome. The estimates for *gag* and *pol* were closer to diagnosis, but still implausible given our knowledge on the recency of the infections (*gag*—median: 141, IQR: 17–266; *pol*—median: 184, IQR: 105–371), whereas estimates for *env* tended to be worse (median: 200, IQR: 25–689).

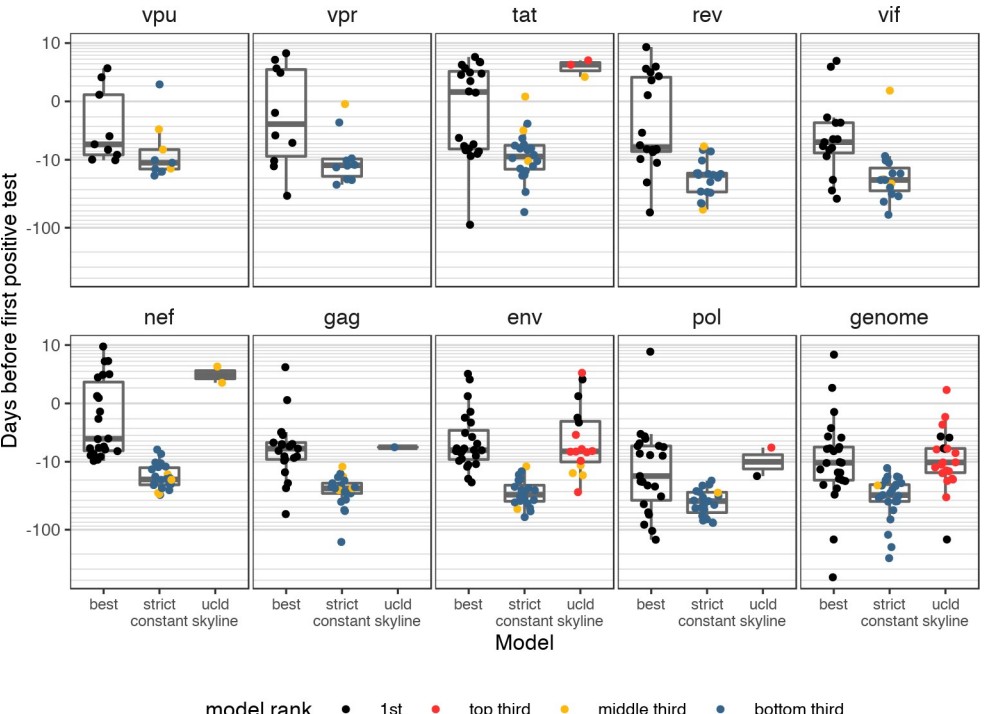

**Fig 6. Estimates of the date of infection for the best-fitting model, compared to the UCLD-skyline and strict-constant models.** Points are colored according to the relative rank of that model out of all models fitted for that participant and gene. Not all participants had a UCLD-skyline or strict-constant model fitted.

We previously split *env* sequences into founder subpopulations, which improved estimates on the whole (IQR: 14 days pre-diagnosis to 1 day post-diagnosis), but gave disparate results between founders [8]. Of the eleven participants infected with multiple founders, seven showed two variants with sufficient phylogenetic signal for estimates of the date of infection to be compared directly. In five participants, the two subpopulations gave estimates of the date of infection that were more than 24 days apart [8]. To see if the increased information in longer sequences could help better resolve estimates, we repeated the analysis using NFL genome sequences, which are over three times longer than *env*.

We used the Gap Procedure [19] on sequences from the initial two time points to identify founders. Five of eleven infections with multiple founders were split into two founder variants and four participants showed three founder lineages; sequences from participants 30812 and 40363 were split into six and five subpopulations respectively. In some participants, the proportion of sequences attributed to each founder were fairly even (e.g. 10203, 10220, 40123), whereas in other participants one variant dominated with >75% majority (e.g. 10463, 20502, 40100, 40436) (Fig 7). After removing founder datasets lacking sufficient information (less than 5 sequences, no temporal signal, or no informative sites), we were left with one founder variant to analyze in BEAST for two participants (20502 and 40100), and two founder variants for the rest.

Overall, results were much improved by splitting the founder populations (Fig 7). Comparing the best-fitting model estimate from each subpopulation versus all sequences combined, there was a median improvement of 228 days (IQR: 86–404), though the range was still wider than the previous *env* analysis (IQR: 20 days pre-diagnosis to 0.051 days post-diagnosis). In four participants, at least one of the estimates for the date of infection was post-diagnosis (10463, 30124, 40363 and 40436), and four participants (10203, 20337, 30124 and 40123) had estimates more than a month before their last negative test. Within a single participant, estimates tended to be disparate, and were more than 30 days apart in five out of nine cases where two subpopulations were analyzed, with very little overlap between the posterior distributions (Fig 7). However, two participants, 10220 and 30812, had concordant estimates between subpopulations (overlap coefficient > 0.7), which fell only a few days prior to diagnosis.

## Discussion

We have used phylogenetic modelling to estimate the date of infection in 39 participants infected with HIV-1, comparing across genes and NFL genome. We showed that the estimated date of infection for single founder infections is around a week prior to diagnosis when using the best-fitting model for *env* and *gag*, whilst the NFL genome placed it around 10 days. Our results emphasized that, in early infection, only the longest genes (*gag, pol, env* and NFL genomes) tended to have sufficient nucleotide substitutions to be amenable for phylogenetic dating, though *pol* tended to give poorer estimates compared to known windows of last negative to first positive HIV-1 RNA test. It remains unclear why the results from *pol* tended to overestimate the date of infection. Previous work has shown that *env* has an elevated evolutionary rate compared to *gag* and *pol*, and that selection acts differently upon the 3' and 5' halves of the genome [20]. The NFL genome results likely reflect the average of these effects between genes, however, given that the estimates from *gag* were similar to those from *env*, these factors alone do not explain the discrepancy with *pol*.

We also highlight that the simplest models, i.e. strict clock, constant population, which are often included as null models in studies, were not typically identified as best-fitting in our cohort (ranking outside the top 5 for all but one participant in *gag* and *pol*), and tend to overestimate the eclipse phase compared to other models. Given the dynamics of viral load in acute

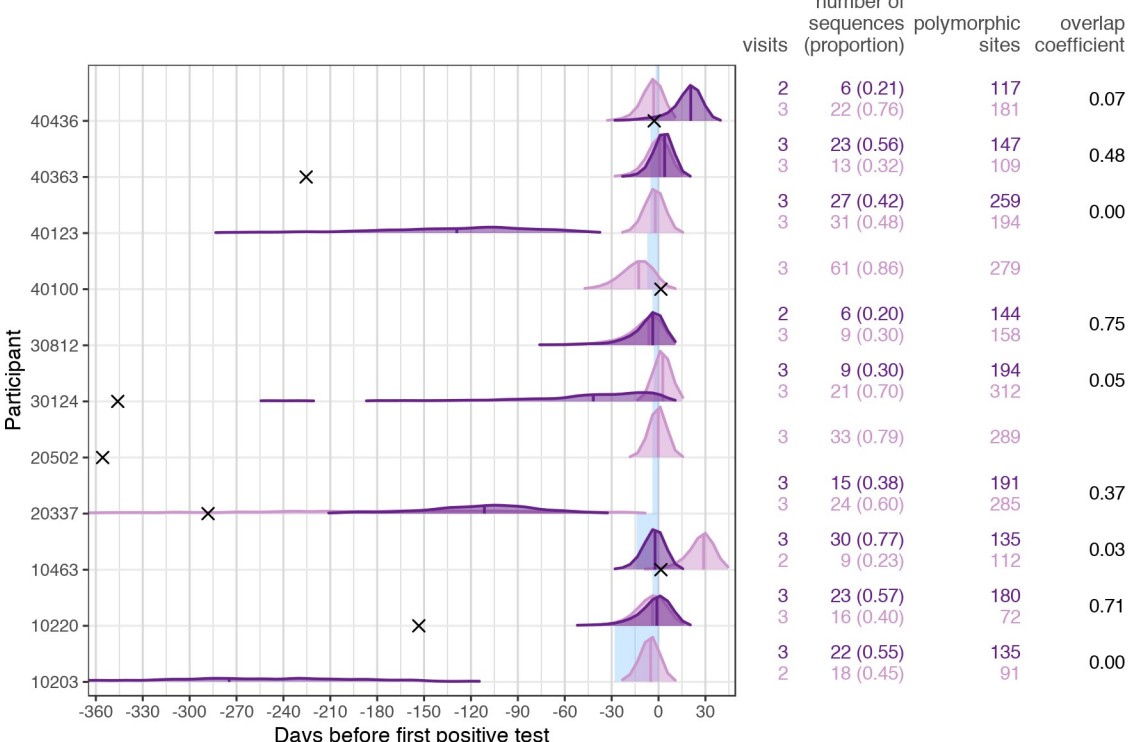

**Fig 7. Improved BEAST estimates on the subpopulations from infections with multiple founder variants for the NFL genome.**
The posterior distributions for the best-fitting model for each identified founder population are shown, with vertical lines marking the median. The shaded blue area corresponds to the interval between the last negative and first positive HIV-1 RNA test (or diagnosis date). Black crosses show the median estimate from assuming a single population (crosses not shown for estimates beyond 365 days prior to diagnosis, which are figured in Fig 3B). The number of visits, sequences and polymorphic sites corresponding to each subpopulation are reported, along with the overlap coefficient for posterior distributions when two subpopulations were analyzed. Only subpopulations with sequences covering at least two time points, a minimum of five sequences with more than one informative site, and significant temporal signal were analyzed.

infection, which increases rapidly, before a steep decline and finally plateauing at the set point, it is not unexpected that more complex population dynamics were favored over the single-parameter constant population size model. In cases where running the full suite of available tree and clock models for model selection is avoided because it can be computationally intensive, our results indicate that it is preferable to use as default the most flexible model, i.e. uncorrelated log-normal relaxed clock and skyline population, which makes the least assumptions.

We found that shared diversity in acute infection is limited; in single founders the majority of the polymorphic sites were singletons, and up to two thirds of participants had insufficient genetic or temporal signal for the shortest genes. Upon nadir viral load about one month after diagnosis (at the second time point of our study), we saw that the sequences remained relatively homogeneous for single founders, with a maximum of 10 parsimony informative sites across the genome, most of which were found in *env*. Mutations accumulate rapidly after this point with evidence of selection; most phylogenies showed one long branch connecting sequences sampled around six months to the previous two time points. This is consistent with previous findings, which identified little evidence of selected mutations in the first month of infection [21–23]. Most genes had fewer than 30 informative sites across the three sampled time points, with more than 60% of participants having no informative sites found in the shortest genes (*vpu* and *vpr*). This is problematic for parameter-rich phylodynamic analyses,

and we found that the lack of signal gave estimates heavily influenced by the prior distribution rather than being biologically plausible (S7 Fig). As an example, we would expect estimates for the evolutionary rate on the order of $10^{-5}$ substitutions per site per day (derived from [24]). However, over one third of the point estimates for *vpr*, *tat* and *rev* were more than 100 times higher than this, and reflected the median of the prior, which had a median of 0.067. Substantial prior posterior overlap, with the prior distribution driving the posterior, suggests that the data is having little influence on the results and this non-identifiability can give misleading results [25]. One way to overcome this could be a more informative prior, although it could also serve to exacerbate the problem and obscure that there was not sufficient information to drive the output. We recommend checking the prior posterior overlap to check for identifiability problems, especially for short genes with low genetic diversity [26].

We found two scenarios where estimated dates were implausible: estimates that occurred after the first positive test and estimates that fell many weeks or months prior to the last negative test. Both are clearly problematic for accurately dating when infection took place, but neither are a methodological issue per se. The date of infection was estimated by the time to most recent common ancestor in the tree. For infections dated after diagnosis, this suggests that either the common ancestor of the sampled sequences existed a considerable time after the point of infection, or, possibly, that the evolutionary rate is being overestimated. In both cases, increasing the sequence sample size by adding more sequences at each time point, or more time points, would help better capture the extant lineages and improve estimates. Over-estimates of the time to most recent common ancestor were most commonly seen for participants infected with multiple founders. The phylogenies for these participants had long deep branches, reflecting evolution both in the current host, and in ancestral branches in the previous host prior to transmission. The longer this previous host has been infected, the more mutations their viral population will have accumulated and the more diverse the variants they will likely pass on. As a way to try and overcome this, we attempted to distinguish founder variants, and re-run the analyses for each variant separately. This improved estimates substantially (Fig 7), though were still often unrealistic compared to the known dates of the last negative and first positive HIV-1 RNA tests. For four participants infected with multiple founders, at least one subpopulation estimate gave dates after diagnosis, suggesting that the common ancestor of those sequences occurred much more recently than the founding infection; given the small sample size (as discussed above), it is likely that this set of sequences represented a subpopulation that diverged recently (from a large viral population corresponding to a chronic infection). Four other participants had estimates more than a month before their last negative test. In the cases where phylogenetic analyses could be conducted on two founder subpopulations, all participants had at least one feasible estimate for the date of infection. These results suggest that with more sequence data available, it should be possible to identify with more confidence the estimated date of infection based on the main founder population.

Besides the small number of sequences per participant, recombination across sequences is also likely to negatively impact estimates. The combination of peak viral load and depletion of target CD4 cells that happens in acute infection means that recombination is likely frequent. While it was not an issue for the homogeneous infections with single founders, we found evidence of recombination across the different populations in infections with multiple founders [27,28]. Recombination can distort branch lengths and cause false signals of exponential growth [29], which could explain the discrepancies between subpopulations (due to the small number of sequences we did not remove recombinants when defining populations, since sufficiently distant recombinant lineages might be expected to cluster into their own subpopulation). Fitting a relaxed clock model may help control these, effectively assuming that a recombination event on a branch is equivalent to that branch having a relatively high mutation

rate relative to branches without recombination events. To reduce the possible effects of recombination, we recommend increasing the sequencing depth at each time point, to obtain a sufficiently large dataset to capture and analyze each founder subpopulation separately after having removed all putative recombinants.

However, even in the absence of recombination, accurately identifying distinct founder subpopulations is not necessarily straightforward. Here we used the Gap Procedure on the sequences from the first two time points to define founder subpopulations [19]. Various other methods for clustering sequences exist, although they are more designed for identifying epidemiologically more-closely linked cases in large, population level databases (see, for example, [30–33]). The current standard for establishing whether an infection was founded by a single or multiple lineage(s) is by coupling the visual inspection of phylogenetic trees and Highlighter plots (www.hiv.lanl.gov) [10] with the analysis of measures of sequence diversity. Recently we showed that the principal eigenvalue of the modified graph Laplacian, which is computed from the distance matrix of the phylogeny, can distinguish between trees from single- or multi- founder infections [11]. Using trees reconstructed in IQ-TREE under the same codon partitions and substitutions as the BEAST analyses, the principal eigenvalue divided participants into whether sequences came from single or multiple founder infections with increased accuracy for longer gene segments, showing that the length of the fragment analyzed is important for founder estimations (Fig 2). It therefore could be used to identify which participant data was more likely to be from a single founder to take forward into the more computationally intensive BEAST analysis.

Here we decided to take the model with the maximum marginal likelihood estimated by stepping stone-sampling as the best-fitting [17]. However, there was often very little difference between the top few models, and there appeared in general to be a balance in overall complexity between the clock and population models. Because the marginal likelihoods are very similar, potential alternative approaches to choosing clock and tree models would be to take the model combination with the fewest parameters [34], or to conduct a model averaging approach [35–37]. Model averaging allows for several models to be included in the same analysis, with results weighted by the time the MCMC spends sampling from each model, perhaps representing a good compromise between number of analyses performed and allowing for model selection. Of course, model selection is also dependent on the models fitted; it is also possible none of the models fitted were ideal. This could have been mitigated by including other flexible models, such as the uncorrelated Gamma relaxed clock, or by using the skyride or skygrid population models, which require less user specification than the skyline model we included here [38–40]. Both the birth-death and coalescent models employed here have some vulnerabilities. The birth-death model is a forwards-in-time model designed for the speciation and extinction of lineages at the population level, and has traditionally been applied to between-host epidemiological models of virus dynamics, rather than within-host. While the birth-death model could potentially offer a useful framework for the transition from a localized infection to a systemic one, we did not evaluate this here. Still, it performed surprisingly well, being selected in the best-fitting model combination for the majority of participants infected with a single founder for *gag* (62%) and typically giving a date of infection within one week prior to diagnosis (median: 5.2; range: 0.16–9.5 days prior to diagnosis). It was selected less frequently for *env* and *pol*, but with similar overall accuracy. This suggests it can be adapted to the within-host setting, but caution is recommended, especially in the selection of priors. It was recently reported that a multitude of alternative birth–death models are equally likely to explain a given time-calibrated phylogeny [41], and results are prone to bias if the sampling process is misspecified [42]. It has been suggested that one way to avoid potential problems is to always consider the full posterior distribution, rather than putting emphasis on point

estimates [43]. However, the full posterior distribution is highly dimensional, which may not be easy to incorporate into clinical trial inference. The deterministic coalescent models are more computationally efficient than birth-death models, being a retrospective look backwards-in-time from the tips, but can struggle to deal with demographic stochasticity [44], which we might expect early in the establishment of infection.

In conclusion, we showed that molecular dating methods can estimate the date of infection of single founders, based on sequences sampled in acute and early infection. For the most accurate estimates, we suggest first identifying whether the infection was established by single- or multiple- founder variants, and then using the longest sequences available, preferably the full genome, *env* or *gag*, to help ensure sufficient temporal and genetic signal. We also recommend using relaxed clock and population models to account for population size changes in acute/early infection and the accompanying substitution rate changes alongside model selection if possible. More broadly, our results emphasize that further studies should be done in parallel across different HIV-1 genes to better understand how the choice of a specific gene may dictate certain conclusions.

## Methods

### Ethics statement

Samples were obtained from the RV217 cohort [8,9]. The protocol was approved by the Walter Reed Army Institute of Research and local ethics review boards: the Makerere University Walter Reed Project, Kampala, Uganda; the Walter Reed Project, Kericho, Kenya; the Mbeya Medical Research Centre, Mbeya, Tanzania; and the Armed Forces Research Institute of Medical Sciences, Bangkok, Thailand. Only adult participants were enrolled. Written informed consent was obtained from all participants. We received Institutional Review Board approval to use the samples and all samples were anonymized.

### Participant selection

Participants from the RV217 prospective cohort were selected for sequencing under the following criteria: 1) recency of the last negative visit to the date of diagnosis; 2) multiple time points available for the viral load upslopes; 3) the length of longitudinal follow up after HIV-1 infection; and 4) sample availability [8]. For the purposes of this study, we also required that sequences were available for three sampling time points (around one week, one month and six months post-diagnosis) for the majority of the genome. For some of the participants, only half genomes were available for one of the sampling dates; in these cases, the genes with less than three time points were dropped from the overall analyses.

### Sequence characterization

HIV-1 near full-length genomes were sequenced from plasma samples following the endpoint dilution strategy outlined in [8]. Hypermutated sequences, as identified using the online Hypermut 2.0 tool (available at https://www.hiv.lanl.gov/content/sequence/HYPERMUT/ hypermut.html [45]), were excluded from further analysis. Sequences were annotated with sampling dates, aligned with Mafft, genes extracted with Gene Cutter from the Los Alamos HIV sequence database (https://www.hiv.lanl.gov/content/sequence/GENE_CUTTER/cutter. html), and alignments manually inspected in Mesquite [46,47]. The number of parsimony informative sites was calculated using the pis function in the phyloch package in R [48,49]. This, the intra-host genetic diversity, visual inspection of Highlighter plots (available at www. hiv.lanl.gov [10]), and results from the modified graph Laplacian were used to define

infections with single or multiple founders as previously described in [8]. In a previous study [22], participants 40061 and 40265 were found to have multiple minority variants at levels ≤3.7% using targeted deep sequencing. However, after removing hypermutant sequences, minority variants were not identified within the single genome amplification (SGA) sequences used in this study, and therefore we classify these participants with the single founders.

The best-fit partitioning schemes and models of evolution were assigned according to the Bayesian Information Criterion (BIC) implemented in PartitionFinder v2.1.1 [50]. For individual genes, codon positions were used as partition subsets; for the full genome, subsets included codon positions, gene, gene overlap and intron regions. The BIC favored partitions and substitution models were then implemented in BEAST.

### Modified graph laplacian

The modified graph Laplacian was obtained using the RPANDA package in R for each participant [51,52]. For each graph we computed the principal (or maximum) eigenvalue, which is shown to positively correlate with the multiplicity of the founder population of HIV-1 infections [11], and calculated thresholds to determine single and multiple founders using the median criterion (the median principal eigenvalue plus 0.5 x standard deviation squared), the jump criterion (the position of the largest discrepancy between consecutive ranked principal eigenvalues), and the partition criterion (clustering on the principal eigenvalue by partitioning around medoids [53]).

### Maximum likelihood trees

Maximum likelihood (ML) trees for each participant were obtained with IQ-TREE [54] using the best partitioning scheme found by PartitionFinder. Polytomies were randomly resolved and zero branch lengths were set to 1 x $10^{-7}$. Temporal signal was assessed following Temp-Est [12]. We first used the rtt function in the ape package in R to root the phylogeny according to the maximum correlation between tip sampling time and distance to root. We then fitted a linear regression of the tip sampling times against the root distances to estimate the mutation rate [55]. Only data sets with a significant positive regression slope (that is, the null hypothesis of $\beta_1 = 0$ was rejected at the 5% level in favor of the alternative, $\beta_1 > 0$, as tested with a one-tailed t-test), were analyzed in BEAST.

### BEAST analyses

BEAST v1.8.3 [14] was used for the joint estimation of the molecular dates, rates of evolution and phylogeny as in [8]. We tested four tree priors–the constant population, exponential growth, and skyline coalescent models and the constant rate birth-death model [56,57]–with four clock models: strict clock, uncorrelated log-normal relaxed clock (UCLD), uncorrelated exponential relaxed clock (UCED) and the random local clock (RLC) [40,58].

The main concern with this dataset was that we were at the lower limit in terms of information content for these datasets given that a majority of sequences were sampled in the first month after diagnosis, especially when considering the short genes in the HIV-1 genome. Therefore, we generally opted for non-informative priors with the rationale that sufficient data will overwhelm a non-ideal choice of prior and still produce reasonable posterior distributions. In this way, we could determine instances when our sequence data were not informative for parameter estimation. The prior for the substitution rate prior was a normal distribution with a mean of 2.24 x $10^{-5}$ substitutions per site per day, with a standard deviation of 0.1 and truncated with bounds of 0 and 1. The mean of this distribution was derived from a within-host estimate in the C2V5 region of the envelope by Lemey et al. [24]. For the constant coalescent

model, a log-normal prior with log(mean) of 0 and log(standard deviation) of 1 was used for the effective population size, and for coalescent exponential and skyline models, a uniform prior between 1 and $1.0 \times 10^{100}$ was used. All other priors were left as the default values given in BEAUti v1.8.3. For the birth-death model we used a uniform prior between 0 and 100,000 for the birth rate, a uniform prior between 0 and 1 for the relative death rate, a uniform prior between 0 and 100 for the rate of sampling through time, and a uniform prior between 1.0 and $1.0 \times 10^{100}$ was used for the time the lineage originated.

BEAST was run for 250 million generations, with a thinning interval of 10,000 iterations. MCMC traces were verified using R and the coda package [49,59]. The first 10% of each run was discarded for burn in as standard, with the effective sample size checked to be above 200 for all parameters. The date of infection was estimated as the root height of the phylogeny; unless otherwise stated, we used the posterior median for point estimates alongside the 95% highest posterior density interval [60]. Plots were generated with ggplot2 in R [61].

## BEAST model selection

Model selection was performed using stepping-stone sampling to assess the relative goodness-of-fit between models based on the marginal likelihood score [17]. Stepping-stone sampling estimates the marginal likelihood by using a set of power posteriors to bridge from the prior to posterior. Following the recommendation by Xie and colleagues [17], and since followed by Baele and colleagues [37,62], we selected the power values, $\beta$, on the path from the prior to posterior using 100 evenly spaced quantiles from the Beta($\alpha$,1.0) distribution with $\alpha = 0.3$. This was implemented in BEAST v1.8.2, with a chain length of 1,000,000 [37,62]. The best-fitting model was chosen as the one with the highest estimated marginal likelihood. Bayes factors (BFs), which calculate the likelihood ratio between hypotheses, were derived to quantify the relative fit between models. Kass and Raftery [18] suggest that a BF>3.2 gives substantial evidence in favor of the alternative hypothesis.

## Identifying subpopulations in multiple founders

For infections with multiple founders, sequences corresponding to different founder variants were also analyzed separately for the genome. For each participant, the Gap Procedure [19] was used to cluster the sequences from the first two timepoints and identify founders. For each of these clusters, a consensus was derived. Sequences from the 6-month time point were added to clusters based on the p-distance from the consensus. Resulting founder datasets that had less than five sequences, sequences from only a single visit date, and/or no parsimony informative sites were excluded from further analysis. The remaining datasets were processed as for the BEAST procedure as above.

## Supporting information

**S1 Table. Summary of the estimated dates of HIV-1 infection for participants infected by single or multiple HIV-1 founders by gene.** n denotes the number of participant datasets. Negative estimates denote days prior to diagnosis.
(DOCX)

**S1 Fig. The accumulation of diversity over time for infections with single founders.** At each sampling time point, we calculated the number of polymorphic sites (sites with at least two alleles) and informative sites (sites with multiple alleles found in at least two sequences) for sequences collected up to and including that time.
(TIF)

**S2 Fig. Breakdown of datasets for the BEAST analysis, by founder type.** Two participants did not have samples for all three time points for *gag* and *pol* and were thus removed for these genes; participant 20368 was removed from the NFL genome analysis due to half genomes only being available. The category 'informative sites' refers to participants whose sequences had at least one informative site; these were then tested for significant temporal signal. Only those with at least one informative site and temporal signal were analyzed in BEAST.
(TIF)

**S3 Fig. Summary of the estimated dates of HIV-1 infection for participants with single or multiple HIV-1 founders by gene.**
(TIF)

**S4 Fig. Comparison of estimates for participants infected with a single founder.** A) Point estimates of the date of infection for participants. The color of the points shows the number of informative sites in the within-host dataset for that participant and gene. Lines link gene and genome results for each participant. B) Boxplot of the overlap coefficient for posterior distributions between pairs of genes for each participant. The coefficient is defined between 0 and 1; 0 implies the curves are non-overlapping, and 1 complete overlap.
(TIF)

**S5 Fig. Comparison of Bayes factors for each model combination across genes and participants with significant temporal signal.** The best-fitting model is shown in red. Shades of orange and yellow give the strength of evidence for the best-fitting model relative to the other model combinations fitted, that is, the darker the color, the smaller the improvement by the best-fitting model. Model combinations shown in gray could not be fitted.
(TIF)

**S6 Fig. Comparison of clock priors for the best-fitting population model.** For each participant, the best-fitting estimate from BEAST is marked by a triangle, with a circle showing the result from the other clock models under the same population model. The shaded blue area corresponded to the interval between the last negative and first positive HIV-1 RNA test; where the blue bar is missing, the sequences were unavailable for that participant.
(TIF)

**S7 Fig. Smaller genes have less signal and are more likely to be influenced by the prior.** The estimated substitution rates for each participant and gene (left) are plotted alongside the prior distribution (right). Horizontal dashed lines show the median and IQR of the prior.
(TIF)

## Acknowledgments

We are indebted to the 3,173 individuals who participated in the RV217 study. We thank the RV217 study team: Jintanat Ananworanich, Sheila Peel, Linda Jagodozinski, Jennifer Malia, Mark Manak, Mark Milazzo, Qun Li, Steve Schech, Julie Dorsey Spitz, Peter Dawson, Prossy Sekiziyivu, Francis Kiweewa, Monica Millard, Doug N. Shaffer, Josphat Kosgei, Joseph Oundo, Nyanda Ntinginya, Cornelia Lueer, Abisai Kisinda, Inge Kroidl, Michael Hoelscher, Arne Kroidl, Rapee Trichavaroj, Siriwat Akapirat, Alex Schuetz, Eugene Kroon, Somchai Sriplienchan, Robert J. O'Connell, Mark DeSouza, Mary Marovich, Edith Swann. We wish to acknowledge former HJF employees who contributed to the HIV-1 sequencing: Daniel Silas, Sandra Mendoza-Guerrero, Adam Bates, Karishma Suchday, Amy Nguyen, Joann Harris, Anjali Bhatnagar, Tara Pinto, Stephanie Melton, Sevan Muhammad, Clinton Ogega, Michelle Lazzaro, Felix Tang, Celina Oropeza, Grace Ibitamuno, Joey Patterson.

## Author Contributions

**Conceptualization:** Bethany Dearlove, Christopher L. Owen, Morgane Rolland.

**Data curation:** Bethany Dearlove, Christopher L. Owen, Yifan Li.

**Formal analysis:** Bethany Dearlove, Christopher L. Owen, Eric Lewitus, Yifan Li, Morgane Rolland.

**Funding acquisition:** Jerome H. Kim, Sandhya Vasan, Robert Gramzinski, Nelson L. Michael, Merlin L. Robb.

**Methodology:** Bethany Dearlove, Christopher L. Owen.

**Project administration:** Leigh Anne Eller.

**Resources:** Sodsai Tovanabutra, Eric Sanders-Buell, Meera Bose, Anne-Marie O'Sullivan, Gustavo Kijak, Shana Miller, Kultida Poltavee, Jenica Lee, Lydia Bonar, Elizabeth Harbolick, Bahar Ahani, Phuc Pham, Hannah Kibuuka, Lucas Maganga, Sorachai Nitayaphan, Fred K. Sawe, Leigh Anne Eller.

**Software:** Bethany Dearlove.

**Supervision:** Morgane Rolland.

**Visualization:** Bethany Dearlove.

**Writing – original draft:** Bethany Dearlove, Morgane Rolland.

**Writing – review & editing:** Bethany Dearlove, Sodsai Tovanabutra, Christopher L. Owen, Eric Lewitus, Yifan Li, Eric Sanders-Buell, Meera Bose, Anne-Marie O'Sullivan, Gustavo Kijak, Shana Miller, Kultida Poltavee, Jenica Lee, Lydia Bonar, Elizabeth Harbolick, Bahar Ahani, Phuc Pham, Hannah Kibuuka, Lucas Maganga, Sorachai Nitayaphan, Fred K. Sawe, Jerome H. Kim, Leigh Anne Eller, Sandhya Vasan, Robert Gramzinski, Nelson L. Michael, Merlin L. Robb, Morgane Rolland.

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
