## [Decision Letter · Decision Letter 0]

5 Jun 2020

Dear Dr Rolland,

Thank you very much for submitting your manuscript "Factors influencing estimates of HIV-1 acute infection timing using BEAST" for consideration at PLOS Computational Biology.

As with all papers reviewed by the journal, your manuscript was reviewed by members of the editorial board and by several independent reviewers. In light of the reviews (below this email), we would like to invite the resubmission of a significantly-revised version that takes into account the reviewers' comments.

We cannot make any decision about publication until we have seen the revised manuscript and your response to the reviewers' comments. Your revised manuscript is also likely to be sent to reviewers for further evaluation.

Sincerely,

Roger Dimitri Kouyos

Associate Editor

PLOS Computational Biology

Rob De Boer

Deputy Editor

PLOS Computational Biology

Reviewer's Responses to Questions

**Comments to the Authors:**

Reviewer #1: Dearlove et al use serially-sampled viral genome sequences from a group of recently-infected HIV patients whose plausible seroconversion dates are known with some precision. They analyse these to determine the level of genetic diversity present in samples at different time points. They then run BEAST with a wide variety of models on each within-host dataset and use model selection to recommend optimal settings. This is a useful study, taking advantage of an unusually comprehensive dataset to make recommendations for general practice. The analyses presented are sensible and well-conducted. I do, however, have a number of substantial concerns. Some concern choices of what to emphasise in the results, others undiscussed limitations.

The manuscript does not actually mention what value it uses as the estimated date of infection (and nor does the accepted Rolland et al. paper). I am assuming, in the absence of a description of any other method, that this is simply the time of the tree root. But it should be noted that this is not necessarily the same thing, and that results that the authors regard as implausible (“infection dates” after diagnosis and differing “infection dates” for multiple founder strains) are in fact perfectly reasonable if the common ancestor of a sample existed some time after the point of infection. TMRCA estimates that are more recent than would be expected from epidemiological data may not really be a problem. This issue certainly should appear in the discussion.

The manuscript also places too much emphasis on point estimates. It’s not really surprising or informative to learn that that posterior medians for the time of infection are not consistent between genes or founder strains. More interesting would be how much overlap there is between the relevant posterior densities.

It feels a little unkind to recommend additional BEAST analyses given the potential time and CPU investment involved, but I do feel that some important models are missing and warrant inclusion. Firstly, the BEAST development team would not have recommended use of the original skyline for many years; the first version with a GMRF smoothing prior appeared in 2008. That the skyride and skygrid are appropriate and preferable in almost all circumstances where the skyline would be used seems to have been poorly communicated to the general community of users. I would be interested to see if results using one of them are meaningfully different, and it would make these results reflect current best practice. Secondly, a relaxed uncorrelated Gamma molecular clock was also added to BEAUti some time ago, and this is a more flexible model than even the lognormal as it can also take a shape with a mode of zero (as with the exponential clock) as well as the peaked shape.

The study has access to enviably detailed and precise data on possible time windows for transmission events, but the manuscript makes surprisingly little use of it. For example, how often does the best-fitting model from stepping-stone sampling estimate an infection date within this window? (Or a credible interval which considerably overlaps it?) Does it ever happen that a less well-fitting model gives a more consistent estimate? Which genes perform best in estimating dates that are consistent with the given intervals?

Finally, while I accept that within-host HIV recombination is a very difficult nut to crack and that it is reasonable to ignore it here, a brief discussion of the problem and how it might have affected the results is warranted.

Minor points

L86: I do not really care for the use of “genome” as a shortening of “whole genome”, even if it is explained here. I have to remind myself what is intended every time I see it.

L167: How was “significant slope” defined?

L191: The authors acknowledge that there was little difference between some of the models, could they not use Bayes factors to quantify this?

L199: Not really a guaranteed overestimate, surely, as you do not know the truth.

L200: When the strict clock was selected as the best model, did it still give earlier estimates than the others?

L220: The pol observation seems quite striking, and I would be interested to see possible explanations in the discussion.

L272: The number of datasets mentioned here do not sum to 11, nor is the change from 7 (L267) explained.

L280: Which BEAST model was used here?

L451: It should be noted that improper priors should not be used in combination with stepping-stone sampling. Were all those replaced with proper ones?

L453: Including the birth-death model at all is a questionable choice. It was, after all, developed as a between-host epidemiological model. I can see how it might be interpreted as a within-host demographic model, but there is also a problem of correlation between the three parameters (see Stadler et al, PNAS, 2013 and in particular the appendix). One of the priors needs to be informative for this analysis to give useful results, and this would have to be carefully chosen if moving the entire framework to a model of a within-host population. This did not happen here.

L463-466: There is a repeated sentence here.

Most of the figures need enlargement of the label text.

Reviewer #2: Overall comments

I was excited to read this paper based on the abstract. But, despite the obviously massive amount of work that went into this paper, I think that the analysis does not make sense and is misleading. I think that the authors need to re-evaluate what their objectives are and re-think what analysis best supports that objective. After reading the paper several basic things are not clear to me. The stated objective seems to be to provide some guidance on how to use multi-sample data (e.g. clones or NGS) to estimate (unknown) infection times, but the analysis seems to be nearly exclusively focused on poorly executed and misleading phylogenetic model selection. I’m also unclear on some basics like how the samples were used, were all the time samples combined into a single tree (a single tree figure would have resoled this). If that is the cases, the authors are analyzing a situation that basically never occurs “in the field” (almost none of the many HIV sequences in the referenced will have multiple clone/NGS from multiple time points); if each time point was considered separately, then the results were not reported by their time strata. Some of the supplemental also seems to suggest that the analysis was strange (env seems to have one of the lower median estimated evolutionary rates—the authors also use substation rate when they mean evolutionary rate). The authors say that they expect evolutionary rates that are 100 fold lower than the median of the priors. This makes no sense

General comments

It’s not at all clear how multiple infection was determined. The reference is to a paper in press and no other mention is made of how this was done or than to say that alignment, phylogeny, and some kind of graph theoretic method was involved.

Figure 2 shows results using methods from citation 10 I think, but citation 8 still seems to be the main citation here (which I cannot read because it’s not public). I’m also not clear where the time dimension went in this figure. The samples came from 4 to 170 days from infection but this does not seem to be reference in the figure (are these just the first sample?).

In general, it’s not clear why the results are not presented as a function of time from infection. The strength of this study is that that time is known.

I don’t understand the point behind the “best-fitting model combination” and I don’t think that the technical execution makes sense. I’ll address the latter point first. Even if we assume that the stone stepping estimator is unbiased and perfect, model selection by marginal likelihoods does not make sense in this case. The marginal likelihood will be highly dependent on the choice of model priors (the ML integrates the product of the likelihood and the prior over the parameter space). Given that some of the model formulations have 10s of parameters, and include ones that have no real principled prior defaults, it’s almost impossible that the authors have set up a situation where every single prior was truly evaluated for what the most reasonable distribution should be for these datasets. And, in the unlikely situation where they did, those values were not reported in full. I think that it’s more likely that they used more-or-less the default priors in BEAST (for unreported values). In this case, this analysis tells us what the default priors of BEAST think about this data. Further, some of the reported values of the priors don’t make sense: the constant population size had a lognormal prior (logmean=0 sdmean=1), which has an expectation of about 1.68, while the exponential population size was uniform over 1 and 1 google. The marginal likelihood calculations therefore assume that the likelihood when the population is 1 billion billion billion is as important as when the population size is 100. To extent the critique, if we told two competent HIV phylogeneticists to do this analysis and to carefully think about the priors, we would get two very different sets of “best” models not because either were wrong, but because the outcome is highly dependent on “reasonable guesses” (e.g. what is a reasonable prior on the exponential growth rate of an ideal population for these data?). This whole analysis does not make sense to me and I believe is highly misleading.

I also find it confusing that the authors seem to want to talk about which models are the best at recovering the known times of infection, but they use instead a (poor in my opinion) model fit statistic. The authors logic is not really stated so I can’t really say, but I suspect that they used marginal likelihoods because they naturally penalize overfitting by naturally spreading out the integral over a larger parameter space in more complex models. However, the whole phylogenetic model is just a nuisance parameter if the only goal is to recover the infection time. If there was a single model that had the lowest bias in the estimated of the infection time but that was only partially identifiable or had a bad marginal likelihood, would that matter? The authors are not trying to do biology but rather to determine the statistical properties of an estimator.

The vertical grid lines in Figure 3 make reading the boxplots hard. This figure also seems to suggest that using more data (whole genome) leads to worse and more variable estimates. It seems that this result alone suggests either the models being used were not sufficient for the whole genome (i.e. not enough heterogeneity comparted to individual gene models) or that recombination is causing a systemic bias towards longer trees.

Specific comments

(L60) This phrasing is weird, should be made clear that the unknown time is the interval from infection to diagnosis. Also, “often” is misleading; there is always a lag between infection and diagnosis.

(L70) I don’t think of PCR as being a serological assay, but this might be ignorance on my part.

(L73) The authors should pick a single term for the time interval between infection and diagnosis and stick to it (in the first 1.5 pages they use both age and date of infection).

(L104) Why are there 3 medians reported? Is this the median time for the first, second, and third samples?

(L106) What does “mainly identified” mean? Subtype A1 was 9/23, do you mean the most common subtype was A1?

(L108) The Thai woman mentioned here is omitted in the intro.

(L115) This seem tautological as sequence alignment (and ostensibly number of polymorphic sites) was used to determine multiplicity of infection.

(L126) How do the authors know which patients were actually multiply infected? Do they just mean that each method was internally consistent.

Reviewer #3: Overall this is a very interesting paper dealing with a very common potential flaw in phylogenetic analysis -- using separate genes that may be more readily available rather than using full genomes that contain more information for meaningful phylogenetic analysis. These kinds of articles are extremely important in particular due to the existing status quo in the field, where the genes with limited information may be used to make conclusions on an epidemiological scale.

The dataset that is being used in this and some preceding papers by the same authors is very unique and useful. Showing the use of computational methods to narrow down the date of suspected infection is a great result that has major potential impact on the healthcare system, helping to control spread of infection in real-time.

However, there are two major concerns about the current version of the manuscript that I mention again later where they occur in text. The first major concern is the fact that per dataset of three sequences per patient and gene/genome the authors ran 16 independent analyses. The amount of analyses on one small dataset is large and this inherently leads to some levels of false positivity (e.g. of models fitting "well" to the data while in fact the fit is not that good). Unfortunately, the authors do not mention how they counteract this or why it is not an issue.

The second major issue is the selection of priors on different parameters. The priors were obviously intended to be as uninformative of the true values as possible (e.g. very flat uniform distributions used for different parameters), however these kinds of distributions are far from uninformative for parameters with extremely low expected values. E.g. in cases when a major part of the posterior distribution lies between 0 and 0.1, whereas the prior sets a weight of 9999.9 on values larger than 0.1. Unsurprisingly, when the amount of information is low, the posterior is very close to the prior.

Overall, I do not think that the authors should redo any analyses in order for this paper to be scientifically useful and a helpful addition to the existing research body. Moreover, rerunning analyses under slightly different assumptions may further exacerbate the multiple testing issue brought up above.

However, at the stage that it is now the manuscript is missing some crucial explanation for justifying the choices made for the analyses. The prior selection is very specific but also extremely generic given the body of knowledge already present on HIV evolution. With the addition of choice justification the manuscript should be ready for publication.

Additionally, I think that it would be very helpful if the authors also published their Beast configuration files so that other researchers could analyse their datasets in a similar manner.

Major comments:

Line 39 and multiple others:

The paper reports IQRs for the posterior values. It would seem more reasonable to (also) report the standard 95% HPDs that most Bayesian analyses report (and in particular Beast analyses) as this covers most of the values of the distribution and can show how precise the estimates are. For example, if the 95% HPD range is extremely wide, the precision of the estimate is low and few conclusions can be drawn from the median value. The reported IQRs are already quite wide so it would be interesting to see the 95% HPDs.

Line 103:

Were the differences in subtype evolution accounted for? E.g. it could be that different subtypes evolve at very different rates, which may bias the analysis results. In particular, when trying to estimate exact timescales good priors on evolution rates can be crucial for precise estimates.

Line 172:

The paper says that there were 9 datasets for a single gene and 25 datasets for full genomes. How is it possible that there were many more full genome sequences (that should include that single gene) than single gene sequences?

Line 176:

One major concern is that small datasets were re-analysed multiple times setting up different models. While a legitimate criterion was used to determine goodness of fit, I am concerned that the multiple analysis runs have introduced false good fits. Could the authors clarify how this was handled or why it is not an issue?

Line 180:

It is not exactly clear how the analyses are set up in Beast. It is my understanding that individual patient data (single gene or genome) was analysed in a single analysis separate from any other patient data. However, there are only 3 sequences per patient, which is incredibly little for reliable parameter inference. How do the authors counteract that? In particular given that the priors on different parameters such as substitution rates are uninformative (in fact, informative of values very different from the expected) and especially since per participant (i.e. per 3 sequences) 16 different model configurations were ran.

Line 320:

This is a very true statement, analyses with little data do end up mainly reconstructing the priors. This is the primary reason for using more data and informative priors. This statement raises once more the question of prior selection described in the manuscript as a lot is already known about HIV phylogenetics and the priors selected here were extremely broad.

Line 324:

Here is an example of the prior explicitly putting emphasis on an unrealistic value. The only conclusion that can be drawn from this would be that there is not enough information in the data for successful inference.

Line 350:

Could it actually be the estimate of the infection time for the infection source for that patient? I.e. if we are estimating the time of coalescence of the founder variants that all come from a single other source, it would span within-host evolution over two patients -- the sampled patient and the source of infection. Of course, there's a possibility of multiple founders in the source too, but the exact same founders would be extremely unlikely to have been passed on to the sampled patient.

Line 452:

The prior choices described here need further justification.

"a uniform prior between 0 and 100,000 for the birth rate" -- This is in fact a prior that puts an incredibly high weight on any value above 1 (99,999 vs 1).

"a uniform prior between 0 and 100 for the rate of sampling through time" -- The sampling proportion and rate in this case are values that are relatively well known for the background population. It would make more sense to set them sensibly based on existing knowledge to not bias the analyses.

"a uniform prior between 1.0 and 1.0 x 10^100 was used for the time the lineage originated" -- in this case as well, the oldest human lineages are from the 20th century at the earliest and the prior puts a lot of weight on values a long time in the past.

Minor comments (typos, typesetting errors, etc):

Line 28: The abstract says "coalescence analyses" whereas in fact the paper presents analyses based on both the coalescent and the birth death models.

Line 181: The word "were" seems to not match the rest of the sentence structure (maybe it is redundant?).

Line 203: Missing a comma after "but when not".

Line 214: "emcompassing"  "encompassing".

Line 271: The authors refer to 5 out of 11 datasets as the majority, which is not the case.

Line 361: "straight forward"  "straightforward".

Line 463: The first sentence of the paragraph is repeated twice.

line 470: The parameter of the Beta distribution is typeset as a blank square.

Figure 1: The grey and the pink are a bit dull and therefore not easy to see.

Figure 2: The figures are too small to be able to read the sample numbers -- the fonts need to be bigger as well as the figures themselves.

Figure 3: The grey and the pink are a bit dull and therefore not easy to see.

Supplementary Figure 3: The plots are unreadable due to the small size.

**Have all data underlying the figures and results presented in the manuscript been provided?**

Reviewer #1: No: They have not been provided in this submission; it is promised that they will follow.

Reviewer #2: Yes

Reviewer #3: Yes

PLOS authors have the option to publish the peer review history of their article (what does this mean?). If published, this will include your full peer review and any attached files.

Reviewer #1: No

Reviewer #2: No

Reviewer #3: Yes: Jūlija Pečerska
---

## [Decision Letter · Decision Letter 1]

11 Sep 2020

Dear Dr Rolland,

Thank you very much for submitting your manuscript "Factors influencing estimates of HIV-1 infection timing using BEAST" for consideration at PLOS Computational Biology. As with all papers reviewed by the journal, your manuscript was reviewed by members of the editorial board and by several independent reviewers. The reviewers appreciated the attention to an important topic. Based on the reviews, we are likely to accept this manuscript for publication, providing that you modify the manuscript according to the review recommendations.

Sincerely,

Roger Dimitri Kouyos

Associate Editor

PLOS Computational Biology

Rob De Boer

Deputy Editor

PLOS Computational Biology

[LINK]

Reviewer's Responses to Questions

**Comments to the Authors:**

Reviewer #1: My concerns have largely been addressed, but I have a few follow-up points:

L104: I did not spot this the first time, but ten sequences from three time points in 39 individuals would be 1170 sequences, not 1280.

L237: The word “skyline” here is extraneous and a little confusing.

L309-310: Could you calculate overlapping coefficients for the dates estimated using different founder populations from the same individual?

L370-371: I think this is not clearly worded enough, perhaps “the common ancestor of the sampled sequences existed a considerable time after the point of infection”? This issue is also not discussed in the context of multiple founder strains, where it is also an explanation for disagreement in estimates.

L393-394: What evidence of recombination was found, and how? Also, since recombination could make phylogenetic inference unreliable, simply adding more sequences will not solve the problem. It would need to be checked for explicitly. Could recombination potentially explain the too-distant TMRCA estimates for some founder strains in 10203, 20337, 30124 and 40123? If single-gene analyses for the same samples did not behave this way I would strongly suspect it. Or are those long branches themselves the evidence?

L427-431: I don’t think this is worded strongly enough. If the authors agree that the use of the birth-death model is inadvisable, I recommend removing it from the manuscript and adjusting the results accordingly.

Figure 4: This looks worse for the authors than I think it needs to. Perhaps add another shaded area representing plausible eclipse periods?

Reviewer #3: The authors have sufficiently addressed the comments and have made significant adjustments to the manuscript to reflect the answers.

With some very minor remaining comments I think that the manuscript is in a good shape for publication.

Minor comments:

Line 109: From what I understand now there are 30 NFL genomes per 39 participants, which should come to 1,170 sequences in total. Where do the other 110 sequences come from? Or is it that there are not exactly 10 NFL genomes per time point per patient but sometimes randomly a few more?

Lines 231-233: This sentence is slightly different from the version in author answers and that small change makes it very confusing. Suggested change:

"All participants had at least one model combination including a skyline in the top three ranked models, with 14 out of 25 participants having the skyline population model for the top three models selected."

to

"All participants had at least one model combination including a skyline among the top three ranked models, while 14 out of 25 participants had the skyline population model for all the top three models selected."

Line 275: What do "better results" mean in in this context? More precise, more accurate or more realistic maybe? Using a different comparator rather that "better" would make this title clearer.

Line 283: Could there be a typo in "17,127 days"? Or did the authors actually get point estimates for infection at 46.5 years prior to diagnosis?

Line 523: Here I would greatly appreciate the adding of a similar comment as was given to my question on priors. Leaving priors on default values is a very dangerous approach which should be discouraged, as mostly the priors make no sense and are only there such that the software would not crash. The authors selected such priors on purpose and I think that stating that extremely clearly will minimise the risk of someone copying these analyses without thinking this over. While this is mentioned in the discussion now, I think that reiterating the reasoning here is very important.

**Have all data underlying the figures and results presented in the manuscript been provided?**

Reviewer #1: Yes

Reviewer #3: Yes

PLOS authors have the option to publish the peer review history of their article (what does this mean?). If published, this will include your full peer review and any attached files.

Reviewer #1: No

Reviewer #3: **Yes: **Jūlija Pečerska
---

## [Decision Letter · Decision Letter 2]

13 Nov 2020

Dear Dr Rolland,

We are pleased to inform you that your manuscript 'Factors influencing estimates of HIV-1 infection timing using BEAST' has been provisionally accepted for publication in PLOS Computational Biology.

Best regards,

Roger Dimitri Kouyos

Associate Editor

PLOS Computational Biology

Rob De Boer

Deputy Editor

PLOS Computational Biology

Reviewer's Responses to Questions

**Comments to the Authors:**

Reviewer #1: My comments have been addressed

**Have all data underlying the figures and results presented in the manuscript been provided?**

Reviewer #1: Yes

PLOS authors have the option to publish the peer review history of their article (what does this mean?). If published, this will include your full peer review and any attached files.

Reviewer #1: No

---

## [Editor Report · Acceptance letter]

25 Jan 2021

PCOMPBIOL-D-19-01987R2 

Factors influencing estimates of HIV-1 infection timing using BEAST

Dear Dr Rolland,

I am pleased to inform you that your manuscript has been formally accepted for publication in PLOS Computational Biology. Your manuscript is now with our production department and you will be notified of the publication date in due course.

With kind regards,

Alice Ellingham
